# IS-SFD: ILLUMINATION SMOOTHNESS AND SEMANTIC-FREQUENCY DENOISING FOR LOW-LIGHT VIDEO ENHANCEMENT

## ABSTRACT

Low-light video enhancement (LLVE) is important for real-world applications where visibility degradation impairs human perception or downstream vision tasks. While zero-reference methods do not need paired image data, they often have flickering problems and struggle to suppress noise while preserving image details. We propose Illumination Smoothness and Semantic-frequency Denoising (IS-SFD), a zero-reference framework for enhancing low-light videos through temporal illumination modeling and denoising guided by semantic and frequency features. To ensure temporal consistency, we introduce a Gated Illumination Estimation Network (GIE-Net) that adaptively fuses multi-frame features by a gating mechanism guided by multiscale similarity of adjacent video frames. For denoising, we design a Semantic-frequency Guided Reflection Denoising Network (SGRD-Net), which combines frequency features from a DWT encoder and semantic features from a frozen CLIP encoder. These features are fused to suppress noise while maintaining structural details in critical areas such as object boundaries. Experiments demonstrate that IS-SFD outperforms existing methods in visual quality and temporal consistency, establishing a new baseline for zero-reference LLVE. The code will be made available upon acceptance of the paper.

## 1 INTRODUCTION

Videos have been widely used in many fields such as autonomous driving, surveillance, and videography. Unluckily, videos captured in low-light environments often suffer from severe degradations such as color distortion, low visibility, and significant noises. These degradations make it challenging for subsequent computer vision tasks in those applications, such as object detection for pedestrians in autonomous driving and segmentation in surveillance videos. Therefore, it is imperative to design robust and effective low-light video enhancement (LLVE) methods.

Existing LLVE approaches can be categorized into two major types: supervised learning methods and zero-reference methods. Supervised methods rely on paired low/normal-light videos for training. However, collecting high-quality video pairs in real-world scenarios is time and resource intensive, which also requires specially designed hardware systems (Wang et al., 2021),(Yu et al., 2024). Although several LLVE datasets have been proposed, they only cover a limited range of scenarios and motion patterns. Moreover, experiment results may not accurately reflect the true performance due to the misalignment between input and ground-truth video frames (Lin et al., 2024).

In contrast, zero-reference methods require no paired data and learn directly from low-light videos. Traditional methods (Guo et al., 2020) for low-light image enhancement (LLIE) include modeling the physical image formation process (e.g., Retinex-based decomposition) or formulating the task as estimating pixel-wise transformation curves. When applied to LLVE, these methods often explicitly adjust the brightness for natural and consistent illumination across video frames. Meanwhile, they further employ temporal and spatial attention mechanisms for feature fusion across video frames to reduce artifacts. However, recent methods still have two major challenges: 1) Inconsistent brightness across video frames due to illumination fluctuations. 2) It is still difficult to improve the performance of noise suppression and detail preservation.

Facing these challenges, we propose an Illumination Smoothness and Semantic-Frequency Denoising (IS-SFD) method for low-light video enhancement. It is a zero-reference framework that enhances low-light videos through temporal illumination modeling and denoising guided by image

semantics and frequency features. IS-SFD has two key modules: Gated-Illumination Estimation Network (GIE-Net), and Semantics-frequency Guided Reflection Denoising Network (SGRD-Net).

For the first challenge, we design a strategy to smooth illumination fluctuations between video frames. Specifically, we utilize optical flow estimation to evaluate the brightness between adjacent video frames based on the principle of brightness constancy. To evaluate the brightness variation, we run a similarity estimation between the multiscale features of the low-light video frames. Compared with traditional similarity estimation, multiscale similarity estimation provides a more robust and comprehensive analysis by capturing fine-grained details and broader contextual information. Then our strategy in GIE-Net adaptively fuses features from illumination maps generated by the last video frame or multiple previous ones based on the similarity. Meanwhile, we consider the temporal distance to the current video frame as a factor to assign weights for different features. For features that are closer to the current video frame, we assign larger weights.

For the second challenge, to suppress noise while preserving fine-grained details, we design the SGRD-Net to integrate noise-invariant semantic features with frequency features. We find that the fused representation enhances noise and retains rich textures and structural details. Specifically, the semantic features extracted from a ResNet-50 image encoder provide high-level guidance to maintain global structural consistency, while the wavelet-based frequency features preserve local textures and show strong robustness to random noise owing to their directional selectivity. SGRD-Net utilizes a frozen CLIP encoder to extract semantic features that preserve the global structure and reduce the negative impact of noise. To get fine textures and local details, we design a Semantic-Frequency Fusion Module (SFFM). This module extracts high-frequency subbands from each video frame using the discrete wavelet transform (DWT) and fuses them with semantic features through a cross-attention mechanism. The Semantic-Frequency (SF) feature effectively combines noise-invariant semantic features with detail-preserving frequency features. Finally, the SF feature is fed into a decoder to reconstruct the clean output image.

Our main contributions are summarized as follows:

- We propose GIE-Net, which generates an accurate illumination map based on multiscale similarity estimation and adaptive feature fusion through a gating mechanism. It significantly improves the temporal consistency of LLVE.

- We propose SGRD-Net, which utilizes semantic and frequency features of video frames to denoise low-light videos. To the best of our knowledge, we are the first to fuse semantic and frequency features for LLVE.

- Extensive experiments on the SDSD and DID datasets show that IS-SFD outperforms state-of-the-art methods in visual quality and temporal stability.

## 2 RELATED WORKS

### 2.1 LOW-LIGHT IMAGE ENHANCEMENT (LLIE)

**Supervised Learning Methods**: Supervised learning methods rely on paired data. Many researchers utilized the Retinex theory to decompose an image into a pair of reflection and illumination components for image enhancement. Previous work (Lore et al., 2017; Wang et al., 2019; Wei et al., 2018) decomposes low-light images into reflection and illumination maps, and then enhances the brightness by adjusting the illumination map. To improve performance under diverse low-light conditions, diffusion-based methods (Jiang et al., 2023; Yi et al., 2023) further model image noise and optimize the difference between predicted noise and true noise for image restoration. Although these methods achieve promising results on certain datasets, they have weak generalization ability for unseen scenarios due to their dependence on specific training data.

**Zero-Reference Learning Methods**: Zero-Reference learning methods typically rely on low-light or normal-light images to learn an enhancement function that adaptively adjusts illumination, contrast, and color distributions, either through self-regularization losses or by incorporating physically inspired priors. Zero-DCE (Guo et al., 2020) first utilized neural networks to predict the parameters of pre-defined curve functions and applied these curves to low-light images for progressive brightness adjustment. However, its pixel-wise curve estimation can lead to over- or underexposure in some regions. To address this, subsequent work (Li et al., 2021b; 2022) optimized the curve formulations and processing speed, producing more accurate and stable brightness adjustments. Nevertheless, they often amplify noises during the illumination adjustment. Zero-IG (Shi et al., 2024) utilized Retinex theory to address this issue by leveraging illumination maps to guide denoising. Despite the

success of zero-reference methods in image enhancement, we can not directly apply them to LLVE since they typically do not consider temporal information between video frames, which may result in flickering in enhanced videos.

## 2.2 LLVE

**Supervised Learning Methods**: Supervised learning methods rely on paired low-light and normal-light video frames for training. For example, StableLLVE (Zhang et al., 2021) simulates dynamic scene motion through optical flow, achieving temporally stable video enhancement with only static images. StableLLVE (Zhang et al., 2021) ensures temporal stability based on optical flow, while Lin et al. (2024) further improves detail and efficiency using wavelet-domain diffusion and cross-scale attention. However, collecting paired low-light video datasets remains challenging, and these datasets often cover limited scenes. The performance of supervised methods depends on the dataset, which leads to poor generalization in diverse real-world scenarios.

**Zero-Reference Learning Methods**: Zero-reference learning methods reduce the reliance on paired low-light videos by leveraging unsupervised or self-supervised cues. LightenFormer (Lv et al., 2023) introduced an unsupervised LLVE method based on a spatio-temporal cooperative attention Transformer. It achieved dynamic range adjustment and long-range spatio-temporal dependency modeling by the S-curve estimation and a refinement network. However, LightenFormer has high computational complexity due to the Transformer-based architecture. SGZ (Zheng & Gupta, 2022) addressed this by adopting a lightweight design with depthwise separable convolutions and unsupervised semantic guidance. However, their limited representation ability often leads to the loss of fine-grained details.

## 3 METHODOLOGY

### 3.1 OVERVIEW OF THE FRAMEWORK

As illustrated in Fig. 1, our framework consists of three main components: a preprocessing module, GIE-Net, and SGRD-Net. The preprocessing module uses optical flow to obtain the warped illumination map and video frame. GIE-Net utilizes either features from previous video frame or the Feature-Similarity (F-S) pairs in the memory to estimate the illumination of the current video frame. SGRD-Net generates the SF feature and uses it to denoise the reflection map. Note that the reflection map is generated based on the estimated illumination map from GIE-Net.

Specifically, we use the optical flow estimation model CEDFlow (Zuo et al., 2024) to extract motion information between adjacent frames $I_t$ and $I_{t-1}$, which generates an optical flow map $f_{(t-1)\to t}$. Then, we apply backward warping to $I_{t-1}$ and illumination map $L_{t-1}$ to generate the warped video frame $I_{(t-1)\to t}$ and the warped illumination map $L_{(t-1)\to t}$ based on the optical flow map. We further infer the illumination map $L_t$ using GIE-Net. A gating mechanism adaptively selects either the feature $F^L_{(t-1)\to t}$ from the warped illumination map $L_{(t-1)\to t}$ combined with the similarity map $S_t$, or the previous $K$ F-S pairs in the memory. Then they are contented with the feature extracted from $I_t$ to estimate $L_t$.

After obtaining $L_t$, we generate the reflection map $R_t$ based on the Retinex theory. However, it still contains a lot of noise due to the intrinsic low-light acquisition process. Thus, we utilize SGRD-Net to denoise it and get the enhanced video frame $\widehat{R}$.

### 3.2 GIE-NET

Existing methods struggle to guarantee the temporal consistency between low-light video frames due to their limited ability to handle inherent illumination fluctuations. For example, moving objects may suddenly block light sources in nighttime outdoor scenes, which leads to brightness flickering in the enhanced video. Fortunately, our multiscale similarity estimation accurately pinpoints the fluctuation area and smooths it based on a gating mechanism.

Optical flow estimation relies on the brightness constancy assumption between consecutive video frames. Thus, we calculate the similarity of $I_t$ and $I_{(t-1)\to t}$ to identify if the illumination has changed. Note that a high similarity indicates a small illumination variance. Meanwhile, we store the features and similarity maps that have small variance in the memory. If the similarity is high, we use $F^L_{(t-1)\to t}$ as a reference to estimate the illuminance of $L_t$. Otherwise, we utilize the F-S pairs to estimate it. We design GIE-Net based on this strategy.

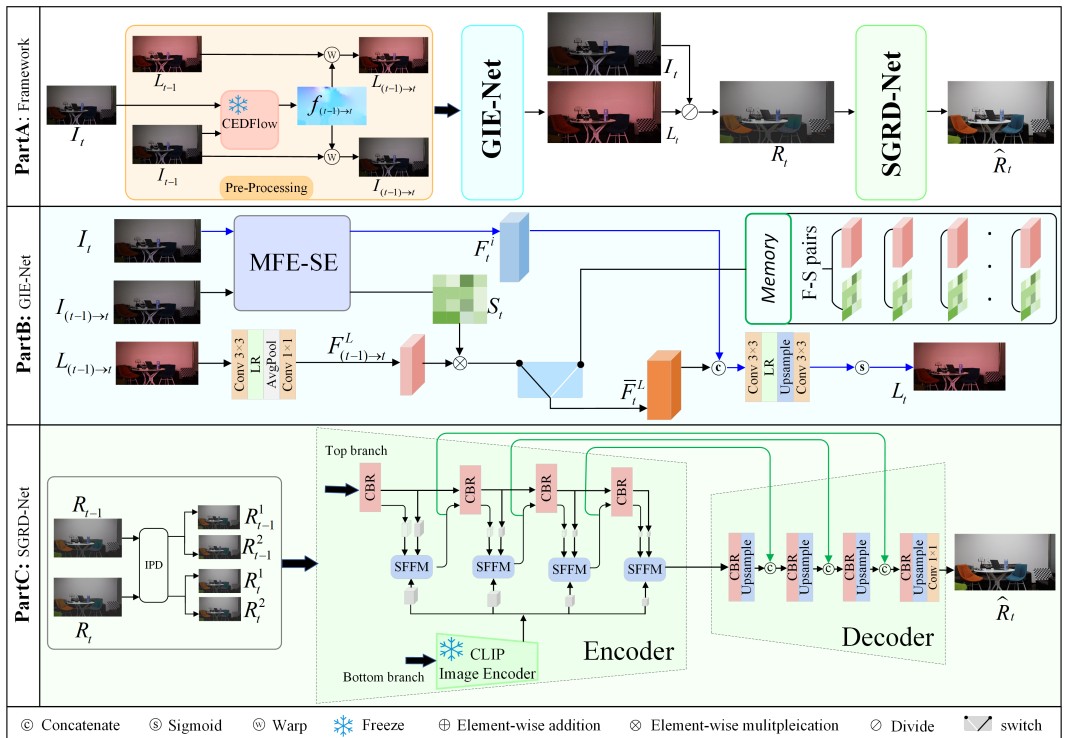

Figure 1: Overall framework of IS-SFD: **Pre-processing** for warped illumination maps and low-light video frames; **GIE-Net** for the illumination map estimation of video frames; **SGRD-Net** for suppressing noise in video frames using semantic and frequency features.

As illustrated in Fig. 1, GIE-NET first obtains the enhanced feature $F_t^i$ and the similarity map $S_t$ between $I_{(t-1)\to t}$ and $I_t$ based on the Multiscale Feature Extraction and Similarity Estimation (MFE-SE) module. MFE-SE preserves more video frame details and structures for better feature representation. It also generates a robust similarity map that reduces errors from local misalignment or visual artifacts. Then, we calculate the score that reflects the illuminance consistency between frames $t$-1 and $t$:

$$score = \frac{1}{HW} \sum_{i=1}^{H} \sum_{j=1}^{W} S_t(i,j). \tag{1}$$

If the score is high, the network selects the $F_{(t-1)\to t}^L$. Otherwise, it utilizes the F-S pairs to compute a smoothed feature representation $\overline{F}_t^L$. This design ensures temporal stability while avoiding feature contamination caused by severely inconsistent illumination. The process is expressed as:

$$\bar{F}_t^L = \begin{cases} \dfrac{\sum_{k=1}^{K} \alpha_k \cdot S_{t-k} \odot F_{(t-k-1)\to(t-k)}^L}{\sum_{k=1}^{K} \alpha_k \cdot S_{t-k} + \epsilon}, & 0 < score < \tau \\[6mm] S_t \odot F_{(t-1)\to t}^L, & score \geq \tau \end{cases} \tag{2}$$

where $\odot$ denotes element-wise multiplication. $\tau$ is a gating threshold. It controls whether to use the $F_{(t-1)\to t}^L$ from the last illumination map or the F-S pairs. A larger $\tau$ requires higher similarity between adjacent video frames. The coefficient $\alpha_k$ adaptively allocates weights for video features in the memory. To emphasize short-term temporal correlations, $\alpha_k$ assigns larger weights for features of video frames closer to the current video frame. $\alpha_k = \frac{\exp(-\lambda k)}{\sum_{j=1}^{K} \exp(-\lambda j)}$, $\lambda > 0$, with $\lambda$ initialized as 1 and set as a learnable parameter during training. $\alpha_k$ is designed to decay exponentially with $k$. A small constant $\epsilon > 0$ is used to avoid numerical instability.

$\overline{F}_t^L$ is then concatenated with $F_t^i$ for illuminance estimation. It is followed by a decoding module composed of $3 \times 3$ Convolution (Conv) + LeakyReLU (LR) + Upsample + $3 \times 3$ Conv, and then a sigmoid function to produce the illumination map $L_t$. For the first video frame, the network directly decodes $F_0^i$ to produce $L_0$ as illustrated by the blue data flow in Fig. 1.

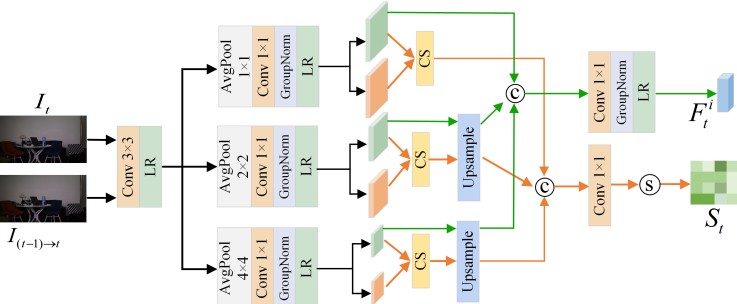

Figure 2: Structure of MFE-SE

**MFE-SE.** MFE-SE is illustrated in Fig. 2. $I_t$ and $I_{(t-1)\to t}$ are first processed separately through $3\times3$ Conv + LR. The resulting feature map is then fed into three parallel branches. Each branch performs an average pooling with $1\times1$, $2\times2$, and $4\times4$ kernel sizes, respectively. The outputs are further processed by $1\times1$ Conv + Group Normalization (GN) + LR.

For image $I_t$, the feature maps from the three branches are concatenated and passed through a fusion module, which consists of $1\times1$ Conv + GN + LR to produce the final enhanced feature $F_t^i$.

To obtain a similarity measure between $I_t$ and $I_{(t-1)\to t}$, we perform multiscale similarity estimation. Specifically, we calculate the cosine similarity (CS) between the features extracted from $I_t$ and $I_{(t-1)\to t}$ for a similarity map in each branch. These maps are then concatenated and processed by a $1\times1$ Conv followed by a sigmoid function to produce a final similarity map $S_t$. To ensure spatial consistency, the features and similarity maps from the 2×2 and 4×4 pooling branches are upsampled to the resolution of the 1×1 branch.

**F-S Pair.** The F-S pairs are stored in a first-in-first-out memory that maintains a sequence of consecutive $K$ most recent pairs, denoted as $\{F_{(t-k-1)\to(t-k)}^L, S_{t-k}, k=1,2,3...K\}$. In our paper, $K = 5$. Each F-S pair includes a feature $F_{(t-k-1)\to(t-k)}^L$ and a similarity map $S_{t-k}$. $F_{(t-k-1)\to(t-k)}^L$ is extracted from the warped illumination map $L_{(t-k-1)\to(t-k)}$ through $3\times3$ Conv + LR + AvgPool + $1\times1$ Conv. For the similarity map, it measures the consistency between $I_{t-k}$ and $I_{(t-k-1)\to(t-k)}$. The memory is initialized with the feature $F_{0\to1}^L$ and similarity map $S_1$ from the first video frame, and updated whenever the similarity score falls below the gating threshold.

### 3.3 SGRD-Net

SGRD-Net suppresses noise in video frames by leveraging a novel SF feature, which is the fusion of semantic and frequency features. The experiment demonstrates that it is noise-invariant for different scenarios and contributes to the noise suppression and detail preservation.

The reflection map $R_t$ is estimated by $I_t$ and $L_t$ based on the Retinex theory. It typically contains significant noise. To reduce the noises, we utilize features extracted by the CLIP image encoder, which are proven to be noise-invariant (Liu et al., 2021). However, they often lack fine-grained structural details since CLIP focuses on high-level semantic abstraction. Thus, we design the SF feature, which integrates semantic and frequency features using SFFM.

SGRD-Net consists of two components: an encoder and a decoder. The encoder comprises four SFFM modules, with each processing features at a distinct scale. The module has three inputs: the semantic feature extracted from CLIP and two image features from a CBR (Conv + BN + ReLU) block. The decoder has four blocks with the same architecture. Each has a CBR block followed by upsampling. Finally, a $1\times1$ convolution layer is applied to produce the denoised reflection image, which is the enhanced low-light video frame.

In SGRD-Net, $R_{t-1}$ and $R_t$ are first fed into an Image Pair Downsampler (IPD) Mansour & Heckel (2023), which generates two downsampled feature maps for each video frame. Specifically, $R_{t-1}^1$ and $R_{t-1}^2$ are generated from $R_{t-1}$, and $R_t^1, R_t^2$ are generated from $R_t$. As illustrated in Fig. 1, the three image pairs $\{R_{t-1}^1, R_t^1\}$, $\{R_{t-1}^2, R_t^2\}$, $\{R_{t-1}, R_t\}$ are processed by a CBR block in the top

branch, respectively. The outputs then serve as sequential inputs to SFFM. For the bottom branch, $R_t$ is fed into a frozen CLIP for the semantic feature $F_s$ that is also inputted to SFFM.

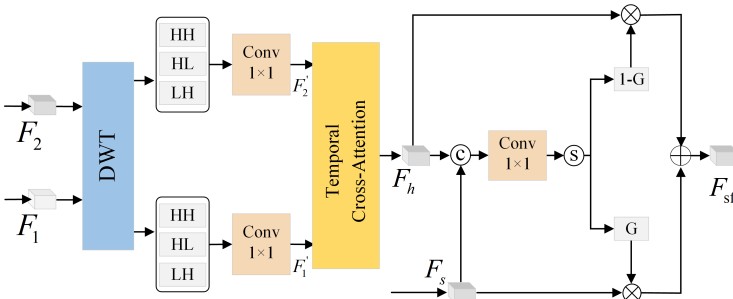

Figure 3: Structure of the SFFM

**SFFM.** To generate SF feature between two adjacent video frames, we propose the SFFM as illustrated in Fig. 3. Given two consecutive video frame features $F_1$ and $F_2$, we first decompose them using DWT to generate three distinct sub-bands: LH (low-high), HL (high-low), and HH (high-high) since they primarily capture fine-grained vertical, horizontal, and diagonal video frame structures. Then they are concatenated and processed by a $1\times1$ Conv layer for channel compression and feature integration. It generates the compact directional representations $F_1'$ and $F_2'$ for the two consecutive video frames. $F_1'$ and $F_2'$ are input to a temporal cross-attention module for the temporal alignment between frames $t$-1 and $t$. Thus, we get the frequency feature $F_h$:

$$F_h = TCA(Q, K, V) = \text{softmax}\left(\frac{QK^T}{\sqrt{d}}\right)V, \tag{3}$$

where $Q = W_q(F_2')$, $K = W_k(F_1')$, and $V = W_v(F_1')$. $W_q, W_k, W_v$ denote learnable linear projections and $d$ is the projection dimension. Subsequently, $F_h$ and $F_s$ are concatenated and passed through a $1\times1$ Conv and sigmoid function for the weight $G$. The outputs are then adaptively integrated through element-wise multiplication and addition:

$$F_{\text{sf}} = G \odot F_s + (1 - G) \odot F_h. \tag{4}$$

The SF feature $F_{\text{sf}}$ is then forwarded to CBR in the next stage and also preserved as a skip connection for the decoder.

## 4 LOSS FUNCTION

### 4.1 LOSS FUNCTION FOR GIE-NET

In GIE-Net, we design a temporal consistency loss to minimize the illuminance difference between adjacent video frames:

$$\mathcal{L}_{\text{tem}} = \left\| S_t \odot \left( F_t^L - F_{(t-1)\to t}^L \right) \right\|_2, \tag{5}$$

Additionally, we employ other loss functions to regularize the training of GIE-Net following Shi et al. (2024): a global adjustment loss $\mathcal{L}_{\text{glo}}$, a pixel-level adjustment loss $\mathcal{L}_{\text{pix}}$, and a smoothness loss $\mathcal{L}_{\text{smo}}$. Their detailed formulations are provided in the supplementary material. Thus, $\mathcal{L}_{\text{RIE}} = \mathcal{L}_{\text{glo}} + \mathcal{L}_{\text{pix}} + \mathcal{L}_{\text{smo}} + \mathcal{L}_{\text{tem}}$.

### 4.2 LOSS FUNCTION FOR SGRD-NET

We design a residual loss $L_{\text{res}}$ and a consistency loss $L_{\text{cons}}$ for SGRD-Net. They denoise the reflection map by mapping one noisy reflection to a different noisy version of the same reflection Mansour & Heckel (2023). It enables the network to estimate the reflection map without ground truth:

$$L_{\text{res}} = \left\| \widehat{R}_t^1 - R_t^2 \right\|_2^2 + \left\| \widehat{R}_t^2 - R_t^1 \right\|_2^2, \tag{6}$$

$$L_{\text{cons}} = \left\| \text{IPD}_1(\widehat{R}_t) - \widehat{R}_t^1 \right\|_2^2 + \left\| \text{IPD}_2(\widehat{R}_t) - \widehat{R}_t^2 \right\|_2^2, \tag{7}$$

where $\widehat{R}_t^1$ and $\widehat{R}_t^2$ are the denoised reflection maps obtained from the input pairs $(R_t^1, R_{t-1}^1)$ and $(R_t^2, R_{t-1}^2)$, respectively. $\text{IPD}_1(\widehat{R}_t)$ and $\text{IPD}_2(\widehat{R}_t)$ denote the two downsampled images obtained by

applying IPD operation to the denoised reflection map $\widehat{R}_t$. Meanwhile, $\widehat{R}_t^1$ and $\widehat{R}_t^2$ are the denoised reflection maps produced by feeding the downsampled image pairs $(R_t^1, R_{t-1}^1)$ and $(R_t^2, R_{t-1}^2)$ into the SGRD-Net. Thus, our loss for SGRD-NET is: $L_{\text{SGRD}} = L_{\text{res}} + L_{\text{cons}}$.

## 5 EXPERIMENT

### 5.1 EXPERIMENTAL SETUP

**Datasets.** We evaluate our method on two benchmark datasets: SDSD (Wang et al., 2021) and DID (Fu et al., 2023). They provide paired low- and normal-light video sequences under different real-world conditions.

**Implementation Details.** We implement our framework with PyTorch on a single RTX 4090 GPU. The model is trained using the Adam optimizer with $\beta_1 = 0.9$, $\beta_2 = 0.999$, and a weight decay of $3\times10^{-4}$. The initial learning rate is set to $3 \times 10^{-4}$. The training is conducted for 50 epochs. During this process, video frames are randomly cropped into $256 \times 256$ patches and the batch size is 4. The pixel values are normalized to the range [0, 1]. In addition, the F-S pairs do not participate in the backpropagation process.

**Evaluation Metrics.** We measure the image quality and temporal stability of video frames. Specifically, we employ Peak Signal-to-Noise Ratio (PSNR) (Hore & Ziou, 2010) and Structural Similarity (SSIM) (Wang et al., 2004) to evaluate the visual quality between the enhanced output and the ground truth. In addition, we use the Learned Perceptual Image Patch Similarity (LPIPS) (Zhang et al., 2018) to evaluate perceptual similarity. It is a no-reference metric based on deep features extracted from pretrained networks. For the temporal stability, we utilize the Average Luminance Variance (ALV) (Li et al., 2021a) and the Mean Absolute Brightness Difference (MABD) (Jiang & Zheng, 2019) for luminance fluctuations.

### 5.2 COMPARISONS WITH STATE-OF-THE-ART METHODS

We compare the IS-SFD with 11 state-of-the-art LLVE/LLIE methods. Note that there are few unsupervised video enhancement methods. Thus, we also compare IS-SFD with five LLIE methods following the setup in Feijoo et al. (2025).

Table 1: Quantitative comparisons on the SDSD dataset. * denotes the LLIE method, and the others are LLVE methods. The best unsupervised methods are in red and the best supervised methods are in blue. For the second-best among these methods, we utilize an underline.

| Methods | | Image quality | | | Video quality | |
|---|---|---|---|---|---|---|
| | | PSNR↑ | SSIM↑ | LPIPS↓ | ALV↓ | MABD↓ |
| Supervised | SMOID(Jiang & Zheng, 2019) | 23.45 | 0.697 | 0.191 | 1.73 | 0.174 |
| | SDSDNet(Wang et al., 2021) | 24.92 | 0.732 | 0.152 | 1.10 | 0.475 |
| | StableLLVE(Zhang et al., 2021) | 20.10 | 0.751 | 0.221 | 1.23 | **0.058** |
| | TCE-Net(Zhu et al., 2024a) | **26.44** | **0.800** | **0.141** | **0.72** | 0.197 |
| | DarkIR*(Feijoo et al., 2025) | 16.99 | 0.640 | 0.222 | 2.89 | 0.161 |
| Unsupervised | Zero-DCE++(Li et al., 2021b) | 13.30 | 0.484 | 0.490 | 9.49 | 0.149 |
| | LightenFormer(Lv et al., 2023) | 20.67 | 0.760 | — | 0.50 | — |
| | Zero-IG*(Shi et al., 2024) | 17.89 | 0.705 | 0.189 | 4.22 | 0.104 |
| | UDU-Net(Zhu et al., 2024b) | 22.41 | 0.736 | — | — | 1.052 |
| | QuadPrior*(Wang et al., 2024) | 16.76 | 0.713 | 0.193 | 4.34 | 0.123 |
| | ULE*(Huaqiu et al., 2025) | 17.60 | 0.623 | 0.265 | 5.09 | 0.345 |
| | IS-SFD(Ours) | 21.58 | **0.772** | **0.185** | **0.47** | **0.087** |

**Evaluation on SDSD.** As shown in Tab. 1, IS-SFD achieves competitive performance on the SDSD dataset. It obtains the best SSIM, LPIPS, ALV and MABD scores among all unsupervised methods. Qualitative results on the test set are presented in Fig. 4. Images generated by DarkIR, ZeroDCE++, and UDU-Net appear too dark, while Zero-IG and ULE show overexposure accompanied by color distortion. Note that we train DarkIR on the SDSD dataset and get the results since the model has not been tested on this dataset yet. StableLLVE produces subtle surface-level noise, appearing as a layer of fine-grained artifacts over the enhanced frames. Additionally, QuadPrior-main loses details in bright regions. For example, only one light spot is correctly enhanced in the zoomed area. In contrast, IS-SFD outperforms other methods in terms of visual quality when compared to the GT.

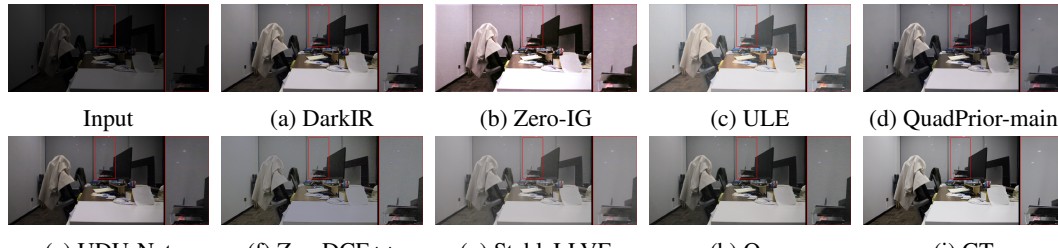

| Input | (a) DarkIR | (b) Zero-IG | (c) ULE | (d) QuadPrior-main |
| (e) UDU-Net | (f) ZeroDCE++ | (g) StableLLVE | (h) Ours | (i) GT |

Figure 4: Results of single-frame comparisons with state-of-the-art methods on the SDSD dataset.

Table 2: Quantitative comparisons on the DID dataset. * denotes LLIE methods. The best results are in **Bold** and the second-best are underlined.

| | Methods | Image quality | | | Video quality | |
|---|---|---|---|---|---|---|
| | | PSNR↑ | SSIM↑ | LPIPS↓ | ALV↓ | MABD↓ |
| Supervised | SMOID(Jiang & Zheng, 2019) | 21.57 | 0.703 | 0.182 | 1.73 | 0.170 |
| | SDSDNet(Wang et al., 2021) | 20.79 | 0.735 | 0.176 | 1.16 | 0.482 |
| | StableLLVE(Zhang et al., 2021) | 19.99 | 0.745 | 0.114 | 0.21 | 0.091 |
| | TCE-Net(Zhu et al., 2024a) | **24.76** | 0.797 | 0.136 | 0.27 | 0.173 |
| | DarkIR*(Feijoo et al., 2025) | 21.87 | 0.796 | 0.130 | 0.10 | 0.104 |
| Unsupervised | Zero-DCE++* (Li et al., 2021b) | 12.67 | 0.611 | 0.240 | 0.24 | 0.224 |
| | Zero-IG* (Shi et al., 2024) | 21.90 | 0.792 | 0.137 | 0.17 | 0.487 |
| | UDU-Net | 20.89 | 0.763 | 0.145 | 0.14 | 1.052 |
| | QuadPrior* (Wang et al., 2024) | 17.31 | 0.737 | 0.184 | 0.13 | 0.128 |
| | ULE* (Huaqiu et al., 2025) | 14.86 | 0.676 | 0.203 | 0.25 | 0.191 |
| | IS-SFD(Ours) | 22.07 | **0.799** | **0.122** | **0.03** | **0.073** |

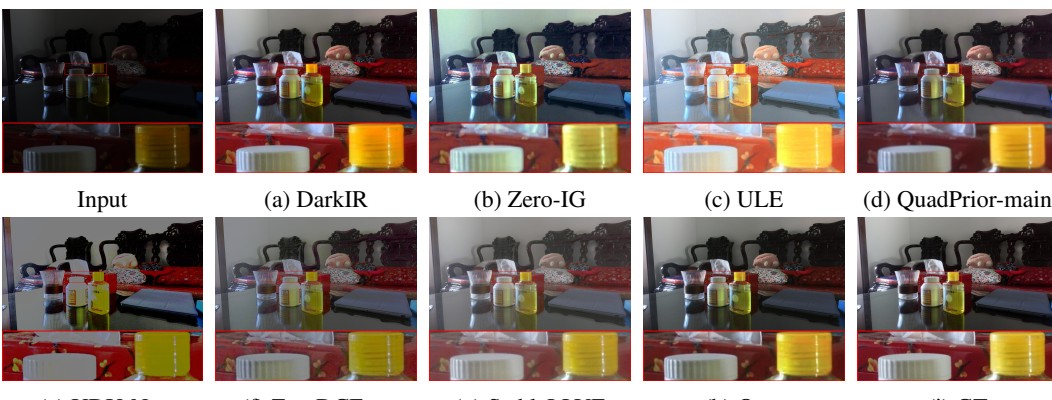

| Input | (a) DarkIR | (b) Zero-IG | (c) ULE | (d) QuadPrior-main |
| (e) UDU-Net | (f) ZeroDCE++ | (g) StableLLVE | (h) Ours | (i) GT |

Figure 5: Results of single-frame comparisons with state-of-the-art methods on the DID dataset.

**Evaluation on DID.** As shown in Tab. 2, IS-SFD achieves the best SSIM, LPIPS, ALV, and MABD and the second-best PSNR on the DID dataset. From the zoomed-in regions of Fig. 5, we find that DarkIR exhibit color distortion, ULE produces overexposure, ZeroDCE++ and StableLLVE produce blurry images with noise. Meanwhile, UDU-Net, QuadPrior-main and Zero-IG lose edge details. In contrast, our method enhances the image brightness while preserving boundary details.

**Evaluation of the multi-frame results.** Fig 6 shows the results reflecting the temporal consistency. We find that SDSDNet produces an unstable light spot across video frames. However, our method achieves smooth brightness. This is because our GIE-Net pays attention to the illumination variance between adjacent video frames.

## 5.3 ABLATION STUDY

**Evaluation of GIE-Net and SGRD-Net.** Tab. 3 shows the performance of GIE-Net and SGRD-Net on the SDSD and DID datasets. We observe that using GIE-Net and SGRD-Net achieves the best performance. For example, PSNR and SSIM are 5.94 dB and 0.149 lower than the best results when we only use GIE-Net on the SDSD dataset. Meanwhile, LPIPS, ALV and MABD increases to 0.271,

2.84 and 1.598, respectively. This is because the illumination map estimation amplifies the noise, which generates a noisy estimated reflection map. If we only use SGRD-Net, PSNR and SSIM are 10.86 dB and 0.279 lower than the best results. This is because the model does not enhance the brightness. Note that we use $I_t$ and $I_{t-1}$ as the input when we only use SGRD-Net.

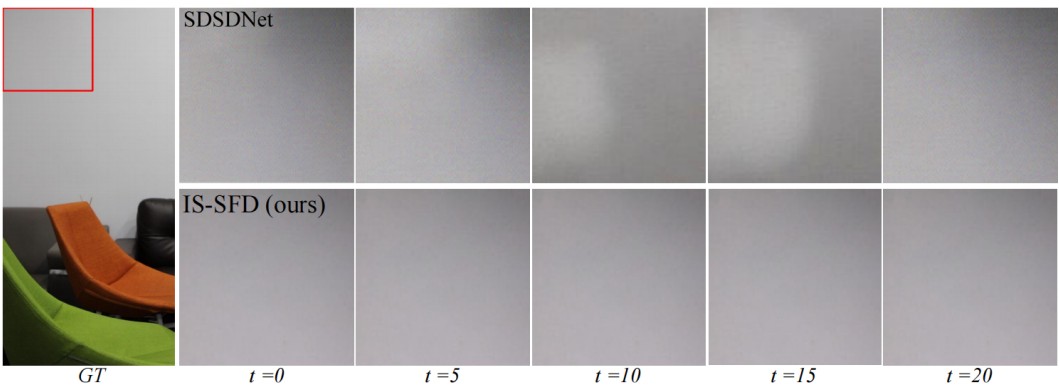

Figure 6: Results of multi-frame comparisons with state-of-the-art methods on the DID dataset.

Table 3: Contribution of GIE-Net and SGRD-Net on different SDSD and DID datasets.

| Dataset | GIE-Net | SGRD-Net | PSNR↑ | SSIM↑ | LPIPS↓ | ALV↓ | MABD↓ |
|---------|---------|----------|-------|-------|--------|------|-------|
| SDSD | ✓ | ✓ | **21.58** | **0.772** | **0.185** | **0.57** | **0.087** |
| | ✓ | ✗ | 15.64 | 0.623 | 0.271 | 2.84 | 1.598 |
| | ✗ | ✓ | 10.72 | 0.493 | 0.451 | 5.52 | 9.101 |
| DID | ✓ | ✓ | **22.07** | **0.799** | **0.122** | **0.30** | **0.071** |
| | ✓ | ✗ | 17.83 | 0.652 | 0.239 | 1.92 | 1.324 |
| | ✗ | ✓ | 11.14 | 0.521 | 0.501 | 5.37 | 8.892 |

**Impact of the Multiscale Similarity.** Tab. 4 shows the impact of different scales. We find that the multiscale similarity map contributes to pinpoint the previous video frame that has small brightness variance compared to current video frame. For example, on the SDSD dataset, a single-scale branch (1×1, 2×2, or 4×4) yields PSNR scores of 21.00, 21.20, and 20.90 dB, respectively. However, our PSNR is 21.58 dB. Meanwhile, SSIM increases from 0.760/0.765/0.755 to 0.772, and LPIPS shows a reduction from 0.190/0.188/0.192 to 0.185. We also find a similar trend on the DID dataset. The results demonstrate the effectiveness of our multiscale similarity map.

Table 4: Impact of the multiscale similarity on SDSD and DID datasets.

| Dataset | Branch | PSNR↑ | SSIM↑ | LPIPS↓ | ALV↓ | MABD↓ |
|---------|--------|-------|-------|--------|------|-------|
| SDSD | 1×1 | 21.00 | 0.760 | 0.190 | 0.60 | 0.090 |
| | 2×2 | 21.20 | 0.765 | 0.188 | 0.59 | 0.088 |
| | 4×4 | 20.90 | 0.755 | 0.192 | 0.61 | 0.092 |
| | IS-SFD(Ours) | **21.58** | **0.772** | **0.185** | **0.57** | **0.087** |
| DID | 1×1 | 21.80 | 0.785 | 0.130 | 0.33 | 0.073 |
| | 2×2 | 21.90 | 0.790 | 0.128 | 0.32 | 0.072 |
| | 4×4 | 21.75 | 0.788 | 0.129 | 0.32 | 0.072 |
| | IS-SFD(Ours) | **22.07** | **0.799** | **0.122** | **0.30** | **0.071** |

## 6 CONCLUSIONS

In this paper, we propose a novel zero-reference method to improve the visual quality and temporal consistency of low-light videos. For temporal consistency, we introduce a strategy for selecting features using an adaptive gating mechanism. Meanwhile, we employ DWT to extract frequency features, and design an SFFM module to fuse them with semantic features from a CLIP image encoder. Thus, we build a SF feature. SF feature contributes to reconstructing clean images due to its noise-invariant property. Compared with existing methods, our approach achieves consistent brightness across video frames and improves the performance of noise suppression and detail preservation. Results demonstrate that our method achieves competitive results on two datasets.

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

## A SUPPLEMENTARY MATERIAL

### A.1 THE NOISE CHALLENGE IN RETINEX-DERIVED REFLECTION MAPS

Images captured in low-light conditions suffer from a weak signal and significant sensor noise. According to Retinex theory, the reflection component $R$ embodies the scene's intrinsic surface properties and should be illumination-invariant. The goal of enhancement is to remove the poor illumination $L$ to retrieve this clear $R$, which would then represent the ideal-enhanced image. A

core challenge, however, is that the decomposition process dramatically amplifies noise. This is because in dark areas, where $L$ is very small, the calculation $R = I/L$ magnifies any noise present in the original image $I$. Thus, the resulting reflection map is often corrupted by severe noise, as demonstrated in Fig. 7.

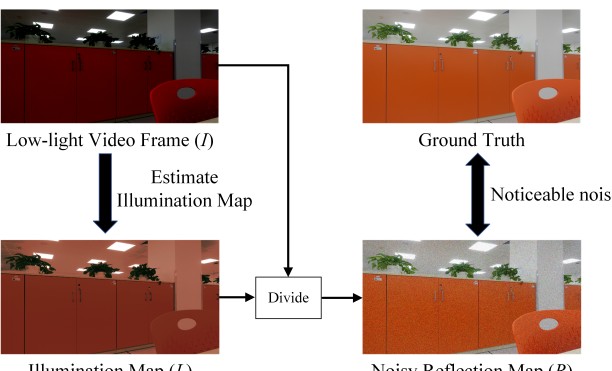

Figure 7: Based on the Retinex theory, we can derive $R = I/L$. In a low-light environment, the video frame $I$ has a significant amount of noise. Thus, the reflection map $R$ is also filled with noise, as shown in the noisy reflection map.

### A.2 Loss function for GIE-Net

The global adjustment loss $\mathcal{L}_{\text{glo}}$ ensures that the brightness of the low-light image is enhanced to match normal-light conditions:

$$\mathcal{L}_{\text{glo}} = \left\| L_t - \alpha^{-1} \right\|_2^2 , \tag{8}$$

where $L_t$ denotes the illumination map of the $t$-th video frame from GIE-Net, and the brightness coefficient $\alpha = \frac{Y_H}{Y_L}$. $Y_H$ denotes the mean luminance of normal-light images, which is 0.5 following Shi et al. (2024). It indicates that video frames are captured under typical illumination with balanced brightness and object details. $Y_L$ refers to the mean luminance of the low-light input image $I_t$.

Uniform enhancement across all pixels may lead to under-enhancement in darker regions or over-exposure in bright areas. Thus, we introduce a constraint on the illumination that adopts pixel-wise intensity. The pixel-level adjustment loss $\mathcal{L}_{\text{pix}}$ is defined as:

$$\mathcal{L}_{\text{pix}} = \left\| L_t - \beta \left( \alpha I_t \right)^\alpha \right\|_2^2 , \tag{9}$$

where $\alpha$ (as in Eq. 8) controls pixel-level scaling, and $\beta$ is a scaling factor that ensures stronger enhancement in darker regions while preventing overexposure in bright ones. $\beta$ is equal to $\alpha^{-1} 0.7^{-\alpha}$ following the setting in Shi et al. (2024).

Furthermore, we introduce a smoothness loss $\mathcal{L}_{\text{smo}}$ for spatial continuity in the illumination map:

$$\mathcal{L}_{\text{smo}} = \sum_{c \in \{R,G,B\}} \left( |\nabla_x L_t^c| + |\nabla_y L_t^c| \right)^2 + \sum_{i=1}^{N} \sum_{j \in \mathcal{N}(i)} w_{i,j} \left| L_t^i - L_t^j \right| , \tag{10}$$

where $\nabla_x$ and $\nabla_y$ represent horizontal and vertical gradient operators, $L_t^c$ denotes the illumination in color channel $c$ of the $t$-th frame, $N$ is the total number of pixels, $\mathcal{N}(i)$ denotes the neighbors of pixel $i$ within a $5\times5$ window, and $w_{i,j}$ is a Gaussian weight computed in YUV space using a Gaussian kernel.

### A.3 Dataset

SDSD is the first public paired low-light video dataset for dynamic scenes. It contains 150 video pairs captured in indoor and outdoor environments using a mechatronic system to ensure uniform motion. Each video consists of 100~300 video frames with a resolution of 1920×1080. We follow the official training/testing split to guarantee fair comparisons.

DID is a large-scale, high-quality dataset designed to address the limitations of existing low-light video datasets, such as limited camera motion, imperfect alignment, and restricted scene diversity. It includes paired low-light and normal-light videos collected from a custom-designed camera system that has accurate spatial alignment for dynamic scenes, even under large camera motion.

DRV (Chen et al., 2019) is a large-scale video dataset for extremely low-light dynamic scenes. It provides raw video sequences captured in indoor and outdoor environments using a Sony RX100 VI camera. Each video contains 50~110 consecutive frames at 20 fps, with a resolution of 3672×5496 in Bayer format.

### A.4 THE NOISE-INVARIANT PROPERTY OF THE SF FEATURE

To evaluate the noise-invariant property of the SF feature, we conduct experiments on three datasets: SDSD, DID, and DRV. We further analyze the stability of their multiscale features under noise perturbations. Specifically, we evaluate the semantic feature $F_s^m$, frequency feature $F_h^m$, and the fused SF feature $F_{\mathrm{sf}}^m$. $m \in \{1, \ldots, 4\}$ denotes the scale index. For each dataset, a normal-light video frame and its corresponding noisy frame generated by GIE-Net are used as inputs. The two frames are forwarded to the frozen SGRD-Net to extract semantic and frequency features at four hierarchical scales. To quantitatively evaluate the noise invariance property, we measure the CS between SF feature extracted at each scale. A higher CS indicates greater feature similarity.

As shown in Fig. 8, the CS remains consistently high across all pyramid levels, indicating the strong noise-invariant ability of the SF feature. The slightly lower CS observed at Scale 1 and 4 can be attributed to the higher susceptibility of shallow and deep frequency features to noise interference, resulting from the spectral overlap between high-frequency noise and fine image details at full resolution and the relative amplification of low-frequency noise artifacts in strongly downsampled representations, respectively. Although the noise robustness of the SF feature may occasionally be inferior to that of purely semantic or frequency-based features, the experimental results in Tab. 5 demonstrate that it ultimately achieves the best overall performance in terms of both image quality (PSNR/SSIM/LPIPS) and temporal consistency (ALV/MABD).

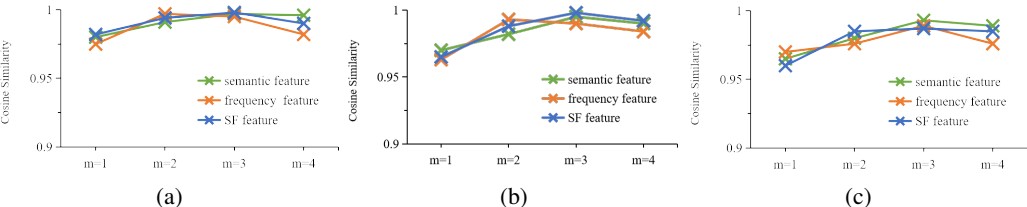

(a)  (b)  (c)

Figure 8: The performance of SF feature. (a), (b), and (c) correspond to the experimental results on the SDSD, DID, and DRV datasets, respectively.

Table 5: Quantitative study of different feature configurations on three datasets.

| Dataset | Semantic | Frequency | PSNR↑ | SSIM↑ | LPIPS↓ | ALV↓ | MABD↓ |
|---------|----------|-----------|-------|-------|--------|------|-------|
| SDSD | ✓ | ✗ | 18.72 | 0.733 | 0.247 | 0.72 | 0.101 |
|  | ✗ | ✓ | 18.12 | 0.677 | 0.228 | 0.66 | 0.089 |
|  | ✓ | ✓ | **21.58** | **0.772** | **0.185** | **0.57** | **0.087** |
| DID | ✓ | ✗ | 20.51 | 0.761 | 0.143 | 0.06 | 0.081 |
|  | ✗ | ✓ | 20.23 | 0.745 | 0.139 | 0.05 | 0.079 |
|  | ✓ | ✓ | **22.07** | **0.799** | **0.122** | **0.03** | **0.070** |
| DRV | ✓ | ✗ | 20.85 | 0.736 | 0.214 | 0.78 | 0.094 |
|  | ✗ | ✓ | 20.42 | 0.722 | 0.207 | 0.75 | 0.092 |
|  | ✓ | ✓ | **21.73** | **0.754** | **0.188** | **0.69** | **0.090** |

### A.5 EVALUATION OF PARAMETER $K$

We investigate the impact of parameter $K$ (the number of F-S pairs) on illumination estimation and temporal stability. Increasing $K$ improves resistance to sudden illumination changes by utilizing a longer temporal history. However, an excessively large $K$ may weaken the influence of recent and highly relevant features. We conduct the experiments on the SDSD test set.

Table 6 shows that increasing $K$ from 3 to 5 consistently improves both image quality (PSNR/SSIM/LPIPS) and temporal consistency (ALV/MABD), with the best results obtained at $K = 5$. This indicates that a moderate number of F-S pairs provides sufficient temporal context without introducing excessive irrelevant information. When $K$ is further increased to 6, 7, or 8, temporal metrics (ALV/MABD) tend to stabilize, but PSNR, SSIM and LPIPS show a mild degradation. This suggests that an excessively large memory prioritizes temporal smoothing at the expense of per-frame fidelity. Therefore, we set $K = 5$ as the default, achieving the best trade-off between single-frame accuracy and temporal consistency.

Table 6: Impact of parameter $K$ on image quality and temporal consistency.

| $K$ | PSNR↑ | SSIM↑ | LPIPS↓ | ALV↓ | MABD↓ |
|---|---|---|---|---|---|
| 3 | 21.20 | 0.758 | 0.205 | 0.66 | 0.103 |
| 4 | 21.40 | 0.765 | 0.194 | 0.61 | 0.095 |
| **5** | **21.58** | **0.772** | **0.185** | **0.57** | **0.087** |
| 6 | 21.52 | 0.770 | 0.183 | 0.58 | 0.088 |
| 7 | 21.45 | 0.767 | 0.191 | 0.58 | 0.087 |
| 8 | 21.35 | 0.763 | 0.198 | 0.58 | 0.088 |

## A.6 EVALUATION OF HYPERPARAMETER $\tau$

We analyze the impact of $\tau$ in Eq. 2 on the illumination map estimation. As shown in Table 7, we observe that $\tau = 0.7$ achieves the best performance considering the trade-off between image quality and temporal consistency. The image quality is low when $\tau$ is small. For example, PSNR and SSIM are 0.74dB and 0.031 less than the best when $\tau$ is 0.5. This is because the model tends to use features from previous video frames even when they are not similar. Thus, it leads to an inaccurate illumination map which introduces extra noise to the estimated reflection map. Meanwhile, the image quality is still not the best when $\tau$ is large although it achieves high temporal consistency. For example, PSNR and SSIM are 0.36dB and 0.018 less than the best and it achieves minor increment in terms of ALV and MABD when $\tau$ is 0.9. This is because the model relies on F-S features due to the strict similarity requirement. Thus, the model neglects the impact of the previous video frame which includes useful features.

Table 7: Impact of hyperparameter $\tau$ on image quality and temporal consistency.

| $\tau$ | SDSD | | | | | DID | | | | |
|---|---|---|---|---|---|---|---|---|---|---|
| | PSNR↑ | SSIM↑ | LPIPS↑ | ALV↓ | MABD↓ | PSNR↑ | SSIM↑ | LPIPS↓ | ALV↓ | MABD↓ |
| 0.5 | 20.84 | 0.741 | 0.236 | 0.71 | 0.122 | 21.35 | 0.771 | 0.168 | 0.05 | 0.11 |
| 0.6 | 21.32 | 0.758 | 0.207 | 0.63 | 0.101 | 21.82 | 0.786 | 0.141 | 0.04 | 0.09 |
| 0.7 | **21.58** | **0.772** | **0.185** | 0.57 | 0.087 | **22.07** | **0.799** | **0.122** | 0.03 | 0.07 |
| 0.8 | 21.46 | 0.766 | 0.192 | 0.54 | 0.082 | 21.95 | 0.793 | 0.129 | 0.03 | 0.06 |
| 0.9 | 21.22 | 0.754 | 0.214 | **0.53** | **0.078** | 21.66 | 0.781 | 0.138 | **0.03** | **0.06** |

