# OpenReview forum: "IS-SFD: Illumination Smoothness and Semantic-frequency Denoising for low-light video enhancement"
_ICLR.cc/2026/Conference — ICLR 2026 Conference Withdrawn Submission_

### Official Review · Reviewer_CkCu · 2025-10-20

**Soundness:** 3
**Presentation:** 2
**Contribution:** 2
**Rating:** 2
**Confidence:** 4

**Summary:**

This paper presents IS-SFD, a zero-reference low-light video enhancement (LLVE) method.
IS-SFD contains two core components:
(1) a Gated-Illumination Estimation Network (GIE-Net), which leverages optical flow and feature-similarity (F-S) mechanism to model temporal illumination consistency;
and (2) a Semantics-frequency Guided Reflection Denoising Network (SGRD-Net), which fuses CLIP-based semantic features with wavelet-based frequency features for denoising.
The method is evaluated on two standard benchmarks and demonstrates compelling performance with existing unsupervised methods.

**Strengths:**

**Clarity and Reproducibility:**
The paper is generally written well and is easy to follow. The commitment to release source code strengthens its practical value and reproducibility.

**Technical Interest:**
Explicitly modeling temporal illumination consistency is important for LLVE. Leveraging the brightness consistency assumption of Optical Flow models to detect the temporal illumination changes seems interesting.

**Empirical Performance:** The method demonstrates compelling quantitative performance with existing unsupervised methods on two benchmarks. Multiple ablation studies are appreciated.

**Weaknesses:**

While the results are promising, there are some major concerns about this work in a compelling narrative for its specific designs and a sufficiently rigorous evaluation to support its claims of novelty.

**Conceptual Novelty and Positioning Require Further Justifications:**

The paper assembles several existing techniques (e.g., optical flow, CLIP, wavelet transforms) but does not adequately justify the specificity of its architectural choices or differentiate from previous methods.

***Optical Flow Design:***
While using optical flow to detect temporally inconsistent illumination appears interesting, the paper fails to argue why it is the best choice. A more convincing narrative could contrast it with other potential cues to establish a well-motivated design choice (e.g., monocular depth, which may be less sensitive to illumination changes). The specific choice of the CEDFlow method also lacks justification against other state-of-the-art optical flow methods.

***GIE-Net Design:***
The MFE-SE module and F-S pairs seem interesting, but are not motivated well by a clear scientific insight. Why is these designs for building spatial coherence and temporal memory better than a standard transformer-based attention mechanism that naturally captures these relationships? The design feels like a complex solution to a problem that may have more elegant alternatives. The authors need to highlight the insight behind them.

***SGRD-Net Design:***
The paper positions itself as "the first to fuse semantic and frequency features for LLVE". However, this claim is invalidated by the existing work [A] that uses CLIP and frequency features for enhancing low-light images. The extension to video may be non-trivial but the core concept is similar. Therefore, the paper needs to shift the claim from "first" to "novel adaptation for video", and clearly articulate what specific designs represent a fundamental novelty over simply applying [A] with temporal smoothing. In addition, the specific choice of CLIP over other popular semantic representations (e.g., CLIP v2, DINO, or SAM) also requires justification.

[A] Low-light Image Enhancement via CLIP-Fourier Guided Wavelet Diffusion, ACM Trans. Multimedia Comput. Commun. Appl., 2025

**Evaluation Rigor is Insufficient:**

***Missing Benchmark and Critical Comparisons:***
The absence of results on the DRV dataset and the limited qualitative comparisons weaken the evaluation quality. For a video enhancement paper, providing video comparisons is essential to demonstrate temporal consistency and the absence of flickering artifacts.

***Incomplete Ablation Studies:***
The current ablations by removing entire modules are too coarse, while fine-grained ablations are required to validate the necessity of each complex component. Below are some suggested alternative baselines for comparisons:
(1) Replace CEDFlow with other flow/depth estimators.
(2) Replace the MFE-SE module with a standard (or adapted) transformer block.
(3) Replace the explicit F-S pair memory with a long-range temporal attention mechanism.
(4) Ablate the internal components of the SFFM.
(5) Evaluate the impact of different semantic representations (e.g., CLIP vs. DINO).
(6) Isolate the contribution of each term in the complex training loss function.

These comparisons can help identify the true contributions from a simple, well-trained network of aggregated capacity.

**Incomplete Literature Review:**
The related work section misses several recent and highly relevant works from top-tier venues (e.g., [B-E] from CVPR 2025). This leaves the impression that this work is not well-positioned within the current state-of-the-art. A thorough update on the literature review is necessary.


[B] Efficient Diffusion as Low Light Enhancer, CVPR 2025
[C] HVI: A New Color Space for Low-light Image Enhancement, CVPR 2025
[D] DarkIR: Robust Low-Light Image Restoration, CVPR 2025
[E] Noise Modeling in One Hour: Minimizing Preparation Efforts for Self-supervised Low-Light RAW Image Denoising, CVPR 2025




**Minor Issues for Correction:**

(1) The term "reflection" should be corrected to "reflectance" for referring to the Retinex theory.

(2) Figures 1-3 are not self-contained. The captions should provide sufficient detail to understand figures without reading the main text.

(3) Although the paper does not emphasize efficiency, it would still be better to provide the runtime cost (e.g., FPS) for a video-based method.

**Justification for Recommendation:**

This paper presents a new method with strong empirical results.
However, the paper currently requires significant strengthening. The method design feels engineered without a strong, underlying scientific insight, and the evaluation does not fully substantiate the claims due to missing benchmark/results and insufficiently granular ablations.
The authors are suggested to reframe their contribution around a clearer, more defensible narrative of novelty and to bolster their experimental evidence with the requested comparisons and analyses.

**Questions:**

(1) What makes optical flow the best choice for this task, compared with other modalities, such as depth estimation (are they less sensitive to brightness changes)?

(2) What makes CEDFlow particularly suitable, against existing optical flow-based approaches?

(3) What specific designs make the MFE-SE module novel against existing feature extraction techniques, e.g., a self-attention mechanism-based one that enforces the spatial consistency?

(4) What is the scientific insight behind an explicit construction of F-S pairs? What makes it better than, e.g., an attention mechanism that enforces temporal correlations?

(5) What are the specific novelties inside the SGRD-Net designs? Is it the temporal cross-attention inside the SFFM?

(6) Are there any new insights to combine frequency and CLIP features? What does the learned weight G represent for fusing CLIP and frequency features?

(7) Will this module benefit from more powerful semantic features, e.g., CLIP v2 features? What makes the CLIP features better than other semantic features, such as DINO features?

---

### Official Review · Reviewer_hsGc · 2025-10-21

**Soundness:** 3
**Presentation:** 2
**Contribution:** 3
**Rating:** 6
**Confidence:** 5

**Summary:**

This paper proposes IS-SFD, a zero‑reference low‑light video enhancement framework composed of two main modules: GIE‑Net (Gated Illumination Estimation Network) for temporally consistent illumination estimation via multiscale similarity and a gated memory of feature–similarity (F‑S) pairs; and SGRD‑Net (Semantic‑frequency Guided Reflection Denoising Network) that denoises Retinex‑derived reflection maps by fusing DWT frequency subbands with semantic features from a frozen CLIP encoder via the SFFM (Semantic‑Frequency Fusion Module).

**Strengths:**

1. Clear problem decomposition and targeted design: separate treatment of temporal illumination smoothness and reflection‑domain denoising is well motivated for the two core LLVE challenges.

2. Combining noise‑robust semantic cues (CLIP) with DWT frequency subbands through cross‑attention (SFFM) is a novel and plausible approach to preserve structure while suppressing noise.

3. Evaluations on two established datasets, quantitative tables, and visual examples (including temporal comparisons) enable reasonable assessment.

**Weaknesses:**

1. The author is suggested to compare the time for inference with several methods, because it is important for practical potentials.

2. The soundness of utilizing clip needs to be further evidenced. Because the sematic feature is sparse and the input videos are noisy, the author is suggested to provide some justification about the rightness of clip feature.

3. There should be some cases displaying the temporal priority of this methods, such as qualified results, etc.

4. This method utilizes CEDFlow for warping. If the light is extremely low, can the flow model work properly? the auther is adviced to display relative visual results. The reviewer also wants to see the generalization of this method in some edge cases.

**Questions:**

Please refer to weaknesses.

---

### Official Review · Reviewer_Ka9H · 2025-10-26

**Soundness:** 2
**Presentation:** 2
**Contribution:** 2
**Rating:** 2
**Confidence:** 4

**Summary:**

This paper proposes a zero-reference low-light video enhancement framework called IS-SFD (Illumination Smoothness and Semantic-Frequency Denoising), which aims to improve visual quality and temporal consistency of videos captured in dark environments. The method contains two main modules: a Gated Illumination Estimation Network (GIE-Net) that smooths illumination variations between frames through multiscale similarity estimation and an adaptive gating mechanism, and a Semantic-Frequency Guided Reflection Denoising Network (SGRD-Net) that fuses semantic features from a frozen CLIP encoder with frequency features from discrete wavelet transforms to suppress noise while preserving structural details. Experiments on SDSD and DID datasets show that IS-SFD achieves competitive performance compared to state-of-the-art supervised and zero-reference methods. However, the overall contribution lacks strong originality, and the paper mainly combines existing techniques such as optical-flow-based illumination smoothing, CLIP feature fusion, and DWT-based denoising into a single framework, making the work appear as a conventional module-stacking approach rather than a fundamentally novel or conceptually creative solution.

**Strengths:**

1.The proposed IS-SFD achieves strong quantitative and qualitative results on multiple benchmark datasets (SDSD and DID), showing clear improvements in PSNR, SSIM, LPIPS, and temporal stability metrics over existing methods.

2.The SGRD-Net successfully combines semantic features from CLIP and frequency features from DWT, enabling better denoising while maintaining fine textures and structural details.

3.The paper is well-written and logically organized.

**Weaknesses:**

1.The overall innovation of the paper is weak. The proposed IS-SFD framework mainly combines existing components, illumination smoothing, optical flow-based similarity estimation, DWT-based frequency analysis, and CLIP-based semantic features without introducing a fundamentally new idea or architecture. The contribution feels incremental and primarily relies on module stacking rather than conceptual innovation.

2.The paper does not clearly explain why the proposed combination of modules is important or how each component contributes to advancing the field. There is little analysis of the theoretical motivation or broader implications of the proposed design choices.

3.The authors do not release the model implementation, training code, or pretrained weights. Without access to these resources, it is difficult to reproduce the reported results or verify the claimed improvements.

4.Although the paper includes quantitative comparisons, it lacks a deeper analysis against recent strong baselines, such as diffusion-based LLVE models or transformer-based zero-reference methods. The absence of more recent competitors limits the strength of the empirical evidence.

5.While ablation studies are presented, they mainly report performance changes numerically without deeper interpretation. The paper does not explore the sensitivity or contribution of key hyperparameters (e.g., τ, K) in practical scenarios or provide qualitative examples to illustrate their impact.

6.The experiments are conducted only on two datasets (SDSD and DID), both of which have similar characteristics. There is no test on unseen or more challenging datasets (e.g., real-world night driving or handheld camera footage), which weakens the claim of generalizability.

7.Some parts of the methodology section are overloaded with equations and technical details, while higher-level intuition is missing. The paper would benefit from more conceptual explanation and better visual illustration of the module interactions.

**Questions:**

1.Could the authors clarify what the core innovation of IS-SFD is beyond the integration of existing components? How does the proposed framework conceptually differ from previous zero-reference LLVE works such as LightenFormer (Lv et al., 2023) or Zero-IG (Shi et al., 2024)? A more explicit theoretical or algorithmic novelty would help strengthen the paper’s originality.

2.The paper introduces GIE-Net and SGRD-Net, but their individual contributions are only shown through simple ablation. Could the authors provide deeper analysis or visualization (e.g., feature maps or temporal consistency heatmaps) to show why each component is effective and how they interact?

3.Do the authors plan to release the code, pretrained models, and training settings? Public access is essential to verify the claimed improvements and to enable fair comparison with future works.

4.The experimental section omits comparisons with more recent or stronger LLVE methods such as diffusion-based or transformer-based approaches. Could the authors include these baselines or at least discuss how IS-SFD would perform relative to them?

5.Have the authors tested the model on more diverse or unseen datasets (e.g., real-world night driving videos, handheld low-light recordings)? Demonstrating robustness under varying illumination and motion patterns would make the method more convincing.

6.The gating threshold (τ) and memory length (K) play key roles in the framework. Could the authors provide more analysis on how these parameters affect stability and visual quality in different lighting conditions?

7.The paper largely focuses on empirical performance. Could the authors provide theoretical insights or analysis explaining why the semantic-frequency fusion improves denoising and temporal consistency beyond empirical observation?

8.It would be helpful if the authors could discuss the computational cost and runtime efficiency of the proposed model compared to lightweight baselines, and suggest potential directions for making the method more efficient for real-time applications.

---

### Official Review · Reviewer_4Pey · 2025-10-27

**Soundness:** 2
**Presentation:** 2
**Contribution:** 2
**Rating:** 4
**Confidence:** 4

**Summary:**

This paper proposes a new low-light video enhancement (LLVE) model, IS-SFD. As a zero-reference LLVE method, it does not requires low-normal light pairs for supervised learning, but it is subjected to the flickering and noise problems. To confront these challenges, the authors propose two modules, a GIE-Net that fuses multi-scale features by a gating mechanism to enhance inter-frames consistency and a SGRD-Net which combines frequency and semantic features to guide denoising.

**Strengths:**

+ This work is well-motivatd. GIE-Net and SGRD-Net are tailored for addressing the inherent challenge of zero-reference LLVE models.
+ The F-S pair design is insightful, which may also applys to other video enhancement tasks.

**Weaknesses:**

- Inconsistent presentation of the best/second results in Table 1 and Table 2. It's better to use a unified rule.
- Evaluation is not comprehensive enough. It's better to evaluate with advanced no-reference image quality assessment and video quality assessment models.
- There are altnerative methods to extract frequency features (e.g., DCT, Laplacian pyramid, etc.). The authors should compare DWT with them.
- Similarly, there are many other vision encoders can be used to extract semnatic features. The authors should compare CLIP with them.
- While this work focus on video enhancement rather than image enhancement. It's better to provide video examples in supplementary material. Failing to do so decredits the promise of this work.

**Questions:**

In the last paragraph of section 1, it states that "the semantic features extracted from a ResNet-50 image encoder provide high-level ....". But actually it uses CLIP model. Is this a typo?

---

### Note · Authors · 2025-12-17

**Comment:**

After discussion among the authors, it has been decided that this paper requires further refinement and will therefore be withdrawn.

**Withdrawal Confirmation:**

I have read and agree with the venue's withdrawal policy on behalf of myself and my co-authors.